# Physicochemical Characterization and Antioxidant Capacity of Açaí (*Euterpe oleracea*) in Ecuadorian Region

**DOI:** 10.3390/foods13193046

**Published:** 2024-09-25

**Authors:** Omar Flor-Unda, Fernanda Guanochanga, Iván Samaniego, Verónica Arias, Bladimir Ortiz, Carmen Rosales, Hector Palacios-Cabrera

**Affiliations:** 1Ingeniería Industrial, Facultad de Ingeniería y Ciencias Aplicadas, Universidad de las Américas, Quito 170125, Ecuador; 2Ingeniería Agroindustrial y de Alimentos, Facultad de Ingeniería y Ciencias Aplicadas, Universidad de las Américas, Quito 170125, Ecuador; fernanda.guanochanga@udla.edu.ec; 3Department of Nutrition and Quality, National Institute of Agricultural Research (INIAP), Panamericana Sur Km. 1, Mejía 170516, Ecuador; ivan.samaniego@iniap.gob.ec (I.S.); veronica.arias@iniap.gob.ec (V.A.); bladimir.ortiz@iniap.gob.ec (B.O.); carmen.rosales@iniap.gob.ec (C.R.); 4Escuela de Nutrición, Facultad de Ciencias de la Salud, Universidad Espíritu Santo, Samborondón 0901952, Ecuador

**Keywords:** Açaí, *Euterpe oleracea*, maturity state, physicochemical characterization, antioxidant capacity

## Abstract

The phytochemical components and antioxidant capacity of Açaí (*Euterpe oleracea*) give it nutritional and bioactive characteristics with anti-cancer and anti-inflammatory properties; it is exported mainly from Brazil to various places worldwide. In Ecuador, the cultivated *Euterpe oleracea* variety has an abundant production that has not been used or studied in depth; because of this, it is relevant to expand the study of this fruit’s phytochemical and antioxidant properties. This paper presents the results of evaluating the concentration of antioxidants and antioxidant activity in different stages of maturation and geographical locations of the Açaí, for which samples obtained in the Ecuadorian provinces of Sucumbíos and Orellana have been evaluated. Antioxidant concentrations were determined with a UV/VIS spectrophotometer at 450–760 nm wavelengths. Antioxidant capacity was determined using the ABTS and FRAP methods. It was evidenced that the values of total polyphenols and total flavonoids decrease with increasing ripening; the opposite effect occurs with total anthocyanins that have a higher concentration in ripe fruits and evidencing an antioxidant capacity that decreases with ripening determined by both methods (FRAP and ABTS).

## 1. Introduction

The Açaí (*Euterpe oleracea*) is a small and globose fruit of 1.5 to 2 cm in diameter native to the Amazon area of Brazil [1], this country being the largest producer of Açaí in the world. South American countries such as Colombia [2], Peru [3], Venezuela [4], Bolivia [5], and Ecuador have Açaí crops [6]; however, this potential has not been efficiently exploited.

In the international environment, the Açaí (*Euterpe oleracea* Mart.) has gained significant recognition in the international market due to its nutritional benefits and its versatility in the food and cosmetics industry [7]. Global demand for Açaí has boosted exports from Brazil, particularly from the state of Pará, which is the world’s largest fruit producer [8]. Companies worldwide use Açaí in products such as energy drinks, food supplements, cosmetics, and ice cream, which has created new market niches and expanded economic opportunities for Brazil [9].

Due to the beneficial nutritional profile of Açaí, which includes high levels of antioxidants, fiber, and healthy fats, it has been used in consumption trends worldwide, making it a superfood that has become popular among health-conscious consumers [10]. This growing demand has fostered greater visibility of Açaí in international markets. It has highlighted the importance of Amazonian products in the global economy, promoting fair trade practices and sustainability in the supply chain. Brazil exports Açaí to the United States, the European Union, Australia, and Japan (Figure 1) [11].

Açaí fruits in the immature state have a green color (E1-10%); in the mid-maturity state, the typical color is yellowish green (E2-50%), and in the ripe state, they are dark purple or purple (E3-100%) [12] (see Figure 1). The consumption of Açaí in several countries has increased considerably in recent years, being considered one of the five fruits with the highest antioxidant potential [13,14,15].

Figure 2 shows a bibliometric visualization of keywords obtained in the scientific literature from studies related to Açaí (*Euterpe oleracea*) applications, which include various topics in human health, animal health, nutritional, and functional aspects.

As shown in Figure 2, studies related to Açaí have been extensively developed due to its antioxidant properties and bioactive components, mainly polyphenols and anthocyanins; in vivo and in vitro studies have shown the strong correlation of phytochemical extracts of the fruit with the reduction in oxidative stress in human and animal cells.

Açaí is a source of flavonoids and polyphenolic compounds used for their high antioxidant activity and ability to destroy cancer cells [16]. The presence of proanthocyanidins and high anthocyanin content has been identified in this fruit, with values 15 to 30 times higher than in red wine, with cyanidin-3-rutinoside and cyanidin-3-glucoside being the most abundant in this fruit [10,17,18,19]. Ref. [20] showed that Açaí extracts have higher antioxidant content than those observed in blueberry and blackberry extracts.

Ecuador, Colombia, Venezuela, French Guiana, and some Caribbean countries have started cultivating Açaí using varieties developed by the Brazilian company Embrapa (Eastern Amazon) and processing equipment technologies used in the state of Pará in Brazil [1].

To market and produce Açaí, it is essential to carry out the phytochemical characterization, the concentration of antioxidants (polyphenols, flavonoids, and anthocyanins), and the antioxidant activity of the bioactive components of the cultivated species [2,21,22,23,24,25,26,27].

In Ecuador, the most important production of Açaí (*Euterpe oleracea*) is centered in the provinces of Esmeraldas, Orellana, and Sucumbíos (where it is called palma). In the Amazonian provinces of Sucumbíos and Orellana, crops of the *Euterpe oleracea* species are found; in their regions, the climatic characteristics are similar to those of the state of Pará in Brazil, which is the main Brazilian exporter of Açaí in the world.

The agro-industrial use of Açaí in Ecuador is not significant, being limited to its use as a living fence [28,29].

Studies have been reported for the species *Euterpe precatoria* cultivated in Esmeraldas (Ecuador), according to [26]. The main variety in the production and marketing of Açaí in South America is *Euterpe oleracea*, a variety that has not been studied in Ecuador at present, which is why this work has been developed.

Currently, no scientific articles study the species *Euterpe oleracea* cultivated in Ecuador, and this species has not been characterized or evaluated in terms of its nutritional and functional characteristics. The objective of this work is to determine physicochemical and nutritional parameters, antioxidant concentration, and antioxidant activity of the Açaí variety (*Euterpe oleracea*) in three stages of maturation (E1-10%, E2-50%, and E3-100%) and two geographical locations in Ecuador (Sucumbíos and Orellana). This work is novel since it provides for the first time reference data on the Açaí (*Euterpe oleracea*) grown in Ecuador and serves as a basis for future studies of marketing feasibility and better use of this fruit. In addition, the values obtained for total anthocyanins and phenols for the same species have been compared in countries of South America: Venezuela, Brazil, Colombia, Peru, Bolivia, and Ecuador.

## 2. Materials and Methods

### 2.1. Material Plant

Samples of 1 kg of Açaí palm fruits (*Euterpe oleracea*) were collected directly from wild crops located in two provinces of the Amazon region in Ecuador. The coordinates of the sampling site are 0°02′49″ N 77°19′22″ W (Sucumbíos) and 0°41′25″ S 77°18′30″ W (Orellana). The regions of Sucumbíos and Orellana were chosen for two fundamental criteria: (1) they are the regions where the Açaí species *Euterpe oleracea* is cultivated in Ecuador, and (2) the climate of the region has characteristics similar to the regions in Brazil (humid tropical climate). Sucumbíos and Orellana are representative regions because Açaí is known and used in traditional practices from the ancestral knowledge of the natives of the region. In addition, the soil is rich and of great diversity, and species are preserved without alterations due to large neighboring cities in a climate that corresponds to a tropical ecosystem with abundant rainfall and warm temperatures throughout the year.

The specific places where the Açaí samples were obtained have different climatic characteristics presented in Table 1. They correspond to the areas of Loreto (403 m altitude) in the Orellana province and Francisco Pizarro (1003 m altitude) in the Sucumbíos province.

Açaí fruits were harvested in three stages of ripeness, the selection of which was associated with the color of the fruit’s coating. The first stage of maturity (E1) represents approximately a 10% change in epicarp color, where fruits have a characteristic green color; the second stage of maturity (E2) refers to Açai fruits with a 50% change in epicarp color, where fruits have a general characteristic color of yellowish green. The third stage (E3) is a 100% change in epicarp color with purple fruits.

Three samples were considered per state of maturity (1 kg for each sample) and for each cultivated region (Sucumbíos and Orellana provinces); a total of 6 samples were analyzed in triplicate (N = 18).

#### 2.1.1. Chemical Reagents

The deionized water required for the test was obtained with a MILLI-Q Academic water purification system (Millipore, São Paulo, Brazil). From the Sigma Aldrich laboratory (St. Louis, MO, USA), the standards for (+), Trolox (6-hydroxy-2,5,7 8-tetramilcroman-2-carboxylic acid), ABTS (2,2-azinobis-3-ethyl-benzothiazoline-6-sulfonic acid), catechin, gallic acid, and cyanidin-3-glucoside chloride were obtained.

#### 2.1.2. Sample Preparation

The Açaí samples collected in each production area were classified according to their maturity and subsequent elimination of impurities by washing under running water and separated into two portions. One portion of the fresh samples underwent physicochemical characterization, while the second portion was packed in airtight bags and stored frozen at a temperature of −20 °C. A freeze-drying process was carried out (temperature of −80 °C and pressure of 0.1 pascals). In a Retsch mill model, ZM 200 (Hann, Germany), with stainless steel mesh (1 mm mesh), the previously freeze-dried and sieved samples were ground to ensure a uniform particle size. Subsequently, the ground samples were protected from light in plastic bottles with airtight lids until the time of analysis.

### 2.2. Methodologies

#### 2.2.1. Physicochemical Analysis

##### Determination of Total Soluble Solids (TSSs)

Total soluble solids were determined using an ATAGO digital refractometer (Tokyo, Japan) following the method cited by [30]; the result was expressed in °Brix. Two drops were placed on the prism of the equipment, and the percentage of soluble solids was observed directly on the display.

##### Titratable Acidity (TA) Determination

In an alkaline solution standardized according to the potentiometric method described in [30], titratable acidity was determined. Thirty grams of fruit pulp was weighed and made up to a volume of 200 mL with distilled water. Then, a 20 mL aliquot was placed in a 25 mL beaker and titrated in a 0.1 N sodium hydroxide solution until pH 8.2 was reached. The results are expressed in grams of citric acid per 100 g of sample.

##### Maturity Index (MI)

The maturity index of the fruit was determined using the relationship (Equation (1)) between the total soluble solids content (°Brix) and the titratable acidity (TA) in each fruit according to the method described in [31], based on the following equation.
(1)MI=TSS/TA
where MI is the maturity index, TSS represents total soluble solids (°Brix), and TA is the titratable acidity value (g citric acid/100 g).

### 2.3. Nutritional Quality Analysis

#### 2.3.1. Ashes

The ash content was determined according to the AOAC 923.03 methodology. One gram of sample was weighed in a 25 mL crucible, and then the sample was calcined for 12 h at a temperature of 500 °C, and a Thermolyne 48000 muffle (Dubuque, IA, USA) was used. The sample is cooled in a desiccator for one hour; the crucible is weighed again, and a difference in weight calculates the ash content. The results are stated in grams of ash per 100 g of dry sample.

#### 2.3.2. Humidity

Moisture determination was performed using the AOAC 925.40 method. It consists of weighing 2 g of sample in aluminum capsules and drying in a convection oven (LabLine Imperial V (Vernon Hills, IL, USA)) at 105 °C for 16 h. After drying, the sample is cooled for 2 h; the sample is weighed again together with the capsule, and the moisture content is calculated by weight difference; the result is expressed as a percentage of moisture.

#### 2.3.3. Protein

The AOAC 2001.11 methodology was used to determine the protein. In a 250 mL digestion tube, 1 g of sample was placed; the copper catalyst in tablets (3.5 g of K_2_SO_4_ and 0.4 g of CuSO_4_.5H_2_O) and 15 mL of concentrated sulfuric acid were added; and the tube was heated for 1 h to 400 °C. After this, the solution is cooled for one hour, and distillation and titration are carried out using a Kjeltec model 8400 FOSS automatic protein analyzer (Hillerod, Denmark). The result is expressed in grams of protein per 100 g of dry matter.

#### 2.3.4. Fat

The AOAC 2003.06 method was used to determine the fat content. In total, 0.5 g of the sample was weighed in stainless steel cups, which were covered with cotton. In the FOSS SoxtecTM 2043 team (Hillerod, Denmark), the cups were placed; when the temperature of the heater reached 130 °C, the thimbles were immersed for 10 min, and after this, the fat extraction reflux was initiated for 30 min. Subsequently, for 10 min, the hexane is recovered. Hexane volatilization is performed when the cups are removed from the equipment and placed in a stove at 105 °C for 1 h. At the end, the samples are placed in a desiccator, cooled, and dried. The results are expressed in grams of fat per 100 g of dry matter.

#### 2.3.5. Total Fiber

The AOAC 978.10 method was used to determine total fiber. It consists of weighing 1 g of sample in a porous glass crucible (100 μm), which is placed in FOSS Fibertec 8000 equipment (Hillerod, Denmark); when the temperature of the equipment reaches 120 °C, acid digestion (1.25% *v*/*v* sulfuric acid solution) is initiated, and sequentially alkaline digestion (1.25% *w*/*v* NaOH solution) occurs, 1 h, for each digestion, respectively. After this, the crucibles are washed with distilled water. Samples with the crucibles are placed at 105 °C for 1 h in a LabLine Imperial V convection oven (Vernon Hills, IL, USA) and then calcined at 500 °C for 8 h; the samples are contained in the crucibles. The samples are placed in a desiccator to cool them, and after cooling, the samples are weighed. The results are expressed in grams of fiber per 100 g of dry sample.

### 2.4. Antioxidant Compound Extraction Process

To extract antioxidant compounds, polyphenols, and flavonoids, 0.3 g of dry sample is placed in 15 mL polyethylene centrifuge tubes. To the tubes, 5 mL of the extraction solution (methanol/water/formic acid, 70/30/0.1 *v*/*v*/*v*/*v*.) was added and homogenized by shaking for 5 min in a Mistral 4600 vortex apparatus (Vernon Hills, IL, USA). The samples were then placed in a Cole-Parmer 8892-MTH ultrasonic bath (Vernon Hills, IL, USA) and centrifuged at 5000 rpm for 10 min. The supernatant was placed in a 25 mL amber balloon, and this extraction procedure was repeated four times continuously. This extraction process was used to quantify the antioxidant capacity of the samples.

#### 2.4.1. Determination of Total Polyphenols

According to the methodology described in [30], total polyphenols were quantified. In a 15 mL tube, 1 mL of a diluted extract aliquot was placed, and 6 mL of distilled water and 1 mL of Folin and Ciocalteu’s reagent were added and allowed to stand for 3 min. After this, 2 mL of 20% sodium carbonate was added and heated at 40 °C for 2 min. A blue chromophore that is a product of the reaction was analyzed in a Shimadzu 2600 UV/VIS spectrophotometer (Tokyo, Japan) at 760 nm. The results were expressed as milligrams of gallic acid equivalent per gram of dry sample (mg GAE/g DW).

#### 2.4.2. Determination of Total Flavonoids

According to the method in [32], the total flavonoid content was evaluated. An aliquot of 1 mL of a diluted extract is taken, and 4 mL of distilled water is added, placed in a 15 mL tube, and homogenized. Subsequently, 0.3 mL of sodium nitrite was added, and the patient waited 5 min. Then, 0.3 mL of aluminum chloride at 10% was placed, and the patient was left to rest for 5 min. After this, 2 mL of NaOH 1N is placed, obtaining a pink chromophore. It is brought to 10 mL with distilled water and analyzed in a Shimadzu 2600 UV/VIS spectrophotometer at a wavelength of 490 nm. Results are expressed as milligrams of catechin equivalent per gram of dry sample (mg CAT/g DW).

#### 2.4.3. Determination of Total Anthocyanins

The determination of the total anthocyanin content was carried out according to the methodology developed in [33]. First, the extraction process was carried out, for which 0.3 g of freeze-dried sample was weighed, and 5 mL of a buffer, pH 1.0, was added and homogenized by vortex agitation Mistral 4600 (Vernon Hills, IL, USA) for 5 min. Subsequently, it was subjected to 10 min in an ultrasonic bath, Cole-Parmer 8892-MTH (Vernon Hills, IL, USA), and centrifuged at 5000 rpm for 10 min, leaving the supernatant (extract) separated from the sedimented solids and transferred to an amber balloon measured at 25 mL. The process was repeated four times to optimize the total extraction of anthocyanins. Finally, the extracts were analyzed in a Shimadzu 2600 UV/VIS spectrophotometer at two wavelengths, 450 nm and 750 nm. The procedure was repeated in the same manner using a pH buffer of 4.5 pH. The results were expressed as mg of cyanidin-3-glucoside per gram of dry samples.

### 2.5. Analysis of Antioxidant Capacity

#### 2.5.1. Determination of Antioxidant Capacity by the 2,2-Azinobis (3-ethyl-benzothiazoline-6-sulfonic acid) (ABTS•+) Decolorization Method

The ABTS methodology used to evaluate the antioxidant capacity is described by [34]. It consists of mixing a 1:1 ratio in an amber flask of a potassium persulfate solution (2.45 mM) with an ABTS-+ solution (7 mM), which is allowed to stand for 16 h. Subsequently, a working ABTS-+ solution is pre-measured for absorbance and diluted in a phosphate buffer to an absorbance of 1.1 + 0.01 at 734 nm. Then, in the UV/VIS spectrophotometer, Shimadzu 2600 (Japan, Tokyo), with a wavelength of 734 nm, the absorbance of the reactions was determined from samples of 200 μL of the diluted extract, added with 3.8 mL of the ABTS-+ working solution, which were left at rest for 45 min before reading in the equipment. The result is expressed as μmol Trolox equivalent per gram of dry sample.

#### 2.5.2. Determination of Antioxidant Capacity by the Ferric Reducing Antioxidant Power (FRAP) Method

The description of the FRAP method is cited by [35]. For this, in a 15 mL tube, 1 mL of the sample; 2.5 mL of a buffer, pH 6.6; and 2.5 mL of a 1% potassium ferrocyanide solution are taken, shaken, and rapidly incubated for 20 min at 50 °C. Subsequently, 2.5 mL of water, 2.5 mL of 10% trichloroacetic acid, and 0.5 mL of 1% FeCl3 are added. At the end of the test, it is homogenized in a vortex and left to rest in a dark environment ideal for the formation of the green complex, ferrous chloride–potassium ferrocyanide, which is analyzed at 700 nm in a UV/VIS Shimadzu 2600 spectrophotometer (Tokyo, Japan). The result is expressed in μmol equivalent of Trolox per gram of dry sample.

### 2.6. Statistical Analysis

For each parameter evaluated in the samples at their respective stage of maturity and from two regions, the results were reported as the mean ± standard deviation (N = 3). The study used a completely randomized design in a 2 × 3 factorial arrangement with three replicates per treatment (N = 18). Variance (ANOVA) was applied to establish the influence of the maturity stage and the two regions on the parameters studied. The Tukey test was used at 5% to establish the differences between the means of each treatment, and the data analysis was performed with Statistica 10.0 for Windows (Statsoft, Paris, France).

## 3. Results

### 3.1. Physicochemical Characterization of the Species *Euterpe oleracea* (Ecuador)

This section presents the characterization of soluble solids (TSSs), titratable acidity (TA), and Maturation Index (TSS/AT) (see Table 2), physicochemical parameters that have an intrinsic correlation between them.

#### Soluble Solids (TSSs), Titratable Acidity (AT), and Maturity Index (MI)

The state of ripeness of the Açaí was determined through sampling and the visual inspection of the color of the fruit coating. Previous studies have shown a relationship between the state of ripeness and the coating color. This is due to the formation of pigments and the variability of organic acids and sugar concentration in the fruit [30,36,37].

Table 2 presents the analysis of the TSS, TA, and MI results, the latter of which is determined by the relationship between the TSS and TA contents. The MI was compared with the coating color of the fruits in their three stages of maturity and percentage of color change (E1-10%, E2-50%, and E3-100%) and harvested in two different provinces in Ecuador (Sucumbíos and Orellana). Based on the results obtained from Table 1, as fruit coloration changes from greenish yellow to purple, it was observed that the sugar content (°Brix) increases, and the acidity decreases regardless of the region of origin of the samples. Samples with a maturity degree of E1-10% (green) increased SS levels from 0.45 ± 0.07 (Sucumbíos) and 0.65 ± 0.07 (Orellana) °Brix to 3.6 ± 0.14 and 2.75 ± 0.07 °Brix, respectively, when their degree of maturity reached E3-100% (purple). The TA values represented in % of citric acid corresponding to the degree of maturity E1-10% (green) decreased from 0.44 ± 0.003 (Sucumbíos) and 0.54 ± 0.003 (Orellana) to 0.26 ± 0.011 and 0.22 ± 0.003%, respectively, when reaching the degree of maturity E3-100% (purple). The results are similar to the study presented by [30,38]; in the fruit ripening process, a decrease in acidity and an increase in soluble solids were observed. It is evident that there is a significant effect of the production region and the degree of maturity of the fruit on the increase in TSS and the decrease in TA; this observation is clear from the ANOVA analysis presented in Table 1 (*p* < 0.05).

### 3.2. Nutritional Quality

Table 3 shows the proximate analysis results of Açai fruits. The results measured in g·100 g^−1^ units showed that the fruit has a high fiber content (40 ± 0.47 to 51.55 ± 0.05), followed by carbohydrates (30.92 ± 0.23 to 53.30 ± 0.27), fat (0.83 ± 0.04 to 12.15 ± 0.11), and protein (4.38 ± 0.58 to 5.36 ± 0.55), and lower ash content (0.12 ± 0.01 to 0.15 ± 0.01). These results agree with those obtained in [39,40], who reported that this exotic native fruit has high total carbohydrate and fiber contents, corroborating the exceptional nutritional value of this fruit.

The results obtained showed that the fat and protein content tends to increase as the maturation process takes place, from the E1-10% stage at 0.83 ± 0.04 and 4.38 ± 0.58 g*100 g^−1^ BS (Sucumbíos) and 1.27 ± 0.04 and 4.43 ± 0.17 g*100 g^−1^ BS (Orellana) to the E3-100% state at 12.15 ± 0.11 and 4.95 ± 0.09 g*100 g^−1^ BS and 8.74 ± 0.18 and 5.36 ± 0.55 g*100 g^−1^ BS, respectively, in the two cultivate regions. The fat and protein content reported in both regions of this study, in the mature state (E3-100%), is higher than the fat and protein contents reported in [41] (6.93 ± 0.70 and 1.90 ± 0.04 g*100 g^−1^ BS) and [42] (7.20 and 1.95 g*100 g^−1^ BS) in the same states of maturity. The statistical analysis of the ANOVA results showed that there is an effect of the state of ripeness on the fat and protein content (*p* < 0.05), which is due to the biochemical changes in the fruit during the ripening process. Similarly, this analysis showed an effect of the production area on fat and protein content (*p* < 0.05 and *p* < 0.01, respectively). On the other hand, the fruits of Açaí have a high fiber content, which varies in the range of 46.46 ± 0.26 g*100 g^−1^ BS (Sucumbíos) and 40.51 ± 0.47 g*100 g^−1^ BS (Orellana) in the E1-10% maturity stage at 51.55 ± 0.05 g*100 g^−1^ BS and 53.10 ± 0.15 g*100 g^−1^ BS, respectively, when it reaches the E3-100% maturity stage. These results show increased fiber during the ripening process, confirming that this Amazonian fruit has a high nutritional and functional value [43]. Similarly, the statistical analysis of the results presented in Table 3 showed an effect of the state of maturity and the production region on fiber content (*p* < 0.05). Figure 3 compares the nutritional values presented in Table 3.

### 3.3. Quantification of Antioxidant Compounds

#### 3.3.1. Total Polyphenols

Table 4 presents the average results of the total polyphenol content in Açaí fruits from two production regions and three stages of maturity. The results showed that the PT content decreases as the maturation process occurs in the samples obtained in the cultivated regions (Orellana and Sucumbíos provinces). Thus, the PT content varies between 33.20± 2.18 and 74.14 ± 3.40 mg GAE/g BS in the province of Orellana, and in Sucumbíos, it varied between 33.20 ± 2.18 and 74.14 ± 3.40 mg GAE/g BS. The highest content of TP was found in the E1-10% maturity stage; in this stage of maturity, the samples of Açaí from the province of Orellana presented the highest content of TP (121.81 mg GAE/g BS). Additionally, it was established that the PT content of Açaí decreases when the ripeness state changes from E1-10% to E3-100%, by 55.22% and 57.72% in the fruits collected in Sucumbíos and Orellana, respectively. It is important to mention that the trend of a decrease in total polyphenols during the maturity of the Açaí fruit in both the states of Orellana and Sucumbíos was also reported in the state of Pará in Brazil in a similar study by [44]; however, the decrease in Açaí PT in Brazil reached approximately 81%.

The ANOVA analysis (Table 4) of the results shows that the TP content is influenced by the effect of the ripening stage (*p* < 0.05). This effect occurs due to the biochemical reactions developed during the ripening process, where an accentuated TP content is observed in the E1-10% stage, and this may be because phenolic acids are more abundant in the early stages of fruit development [30,45]. The ANOVA analysis showed that the geographical region of production affects the PT content (*p* < 0.05). The Açaí fruits from the Orellana province in the E1-10% ripening stage and the E3-100% ripe stage have higher PT contents (121.81 ± 2.55 and 51.50 ± 2.17 mg GAE/g BS, respectively) concerning those obtained in the samples from the province of Sucumbíos (74.14 ± 3.40 and 33.20 ± 2.18 mg GAE/g BS).

Table 4 shows that the PT content was 51.50 ± 2.17 (Sucumbíos) and 33.20 ± 2.18 mg GAE/g BS (Orellana) for the maturity stage (E3-100%), which are equivalent to the corresponding units (5.15 ± 0.22 and 3.32 ± 0.22 g GAE/100 g BS), respectively. The results of PT obtained for the Açaí in the Orellana province are higher in the range of approximately 26% to 50% than those presented by the authors of [46,47,48], who reported that the Açaí fruits from the Amazon region of Brazil, Peru, and Colombia have PT contents of 3.8 g GAE/100 g, 2.6 g GAE/100 g, and 3.2 g GAE/100 g, respectively. Ref. [49] reported PT contents in the range of 5.02 ± 0.10 to 2.20 ± 0.20 g GAE/100 g in Açaí fruits from the Amazon region of Venezuela.

#### 3.3.2. Total Flavonoids (FT)

The Açaí fruits showed FT contents that decreased during the ripening process observed (Table 4) in samples from the cultivated regions (Orellana and Sucumbíos provinces). The contents of FT ranged from 42.32± 1.29 to 33.03 ± 0.39 mg catechin/g BS in samples from the province of Orellana, while in Sucumbíos, it ranged from 60.14 ±2.28 to 28.84 ± 3.19 mg catechin/g BS. The results allowed us to establish that the fruits of Açaí in the E1-10% ripe state have a higher content of FT. At maturity stage E1-10%, samples from the province of Orellana have the highest FT content (60.14 ± 2.28 mg catechin/g BS).

The results of total flavonoids expressed in mg/100 g found in our study in the provinces of Orellana and Sucumbíos were 330 and 288.4, respectively, being higher in a range of 26% to 36%, compared to the mean values of eight samples (213 mg/100 g) found in Belén de Pará, Brazil, the traditional region of the production and consumption of Açaí [50].

At full maturity (E3-100%), FT contents were 33.03 ± 0.39 and 28.84 ± 3.19 mg Catechin/g BS for Açaí obtained from the provinces of Orellana and Sucumbíos, respectively (Table 4). The statistical analysis of the ANOVA results showed that there is an effect of the maturity status and the geographical area of production on the FT content (*p* < 0.05).

#### 3.3.3. Total Anthocyanins (ACT)

The content of total anthocyanins in the Açaí fruit is presented in Table 3. It is observed that there is an increase in TCA during the fruit’s ripening process. For the Açaí samples from the Orellana province, the TCA content increases in the maturity range (E1-10% to E3-100%) from 4.28± 0.78 to 99.59 ± 0.65 mg cyanidin-3-glucoside. g-1 BS and for the Sucumbíos samples from 2.86 ± 0.25 to 90.16 ± 1.53 mg cyanidin-3-glucoside. g-1 BS. The decrease in TCA during maturity is in the range of approximately 96%-97% in both regions. The ANOVA analysis of the results (Table 4) established that there is an effect of the maturity stage on TCA content (*p* < 0.05). The results obtained for TCA in the provinces of Orellana and Sucumbíos at the edible maturity stage E3-100% (99.59 ± 0.65 mg cyanidin-3-glucoside. g-1 BS and 90.16 ± 1.53 mg cyanidin-3-glucoside. g-1 BS), respectively, are higher in a range of 53% to 58% than those obtained by [51] in Açaí from Brazil (42.3 mg cyanidin-3-glucoside. g-1 BS). Refs. [52,53] also reported total anthocyanin concentrations in Açaí (54.24 mg cyanidin-3-glucoside. g-1 BS and 18.9 mg cyanidin-3-glucoside. g-1 BS). According to the ANOVA statistical analysis, TCA content is influenced by the geographical area of production (*p* < 0.05).

Figure 4 shows the trend in the behavior of polyphenols, flavonoids, and anthocyanins from Açaí in three different stages of maturation and the provinces of Sucumbíos and Orellana (Ecuador). A decrease in total phenols and flavonoids is observed as the fruit matures, and a contrary trend is observed in the concentration of anthocyanins. The conversion of anthocyanins from non-anthocyanin phenolic precursors influences the increase in anthocyanin content at the ripening stage. The results show that the content of flavonoids and phenols in green fruits is high, which may have potential applications in food and pharmaceutical products [54,55,56]. The decrease in phenolic compounds during the maturation process presented in this study (Figure 4) was also reported in a similar species [57].

### 3.4. Evaluation of Antioxidant Activity by the ABTS Method

The results obtained in the evaluation of the AA by the ABTS method in Açai fruits from two production areas in the regions of Sucumbíos and Orellana in different stages of maturity are presented in Table 5. The results suggest that the AA value decreases from the ripening state E1-10% to E3-100% with values of 1336.96 ± 89.25 μmol trolox·g^−1^ (Sucumbíos) and 1474.97 ± 35.54 μmol trolox·g^−1^ (Orellana) up to 402.41 ± 10.38 and 463.22± 24.92 μmol trolox·g^−1^, respectively. The decrease in AA is 69.90% for the samples from Sucumbíos and 68.59% for the samples from Orellana. At maturity E3-100%, samples of Açaí from the province of Orellana show the highest antioxidant activity.

According to the statistical analysis, the AA values are influenced by the geographic region and maturity grades of the Açaí crop (*p* < 0.05), confirming a decrease during the ripening process in the Açaí samples from the two regions. These results are similar to those cited by [30,58], who demonstrated in studies conducted on blackberry cultivars that AA decreases as the ripening process progresses. The antioxidant activity values found by the ABTS method in samples from the provinces of Sucumbíos and Orellana in eastern Ecuador were seven times higher than the antioxidant capacity found in the state of Pará in Brazil [59].

### 3.5. Evaluation of Antioxidant Activity by the FRAP Method

The results obtained in the evaluation of AA by the FRAP method presented in Table 5 show a decrease in the AA value of the degree of maturity of Açaí from E1-10% to E3-100% with values of 880.58 ± 19.69 μmol trolox·.g^−1^ (Sucumbíos) and 1033.87 ± 19.98 μmol trolox.g^−1^ (Orellana) up to 315.43 ± 4.96 and 430.94 ± 9.23 μmol trolox.g^−1^, respectively. The decrease in the AA value is 64.90% for samples from Sucumbíos and 58.31% for samples from Orellana. In the E3-100% maturity stage, the samples of Açaí from the province of Orellana show the highest antioxidant activity with 430.94 ± 9.23 μmol trolox.g^−1^. The AA values for Sucumbíos and Orellana, according to the FRAP method, are 60% and 87% higher than the similar species grown in Belén de Pará in Brazil [60], while the values obtained by the ABTS method are 70% and 88.1% higher.

In [30] regarding blackberry cultivars, the AA decreases as the ripening process progresses (25.33% and 36.21% to 54.44% decrease, respectively). The antioxidant capacity values determined by the FRAP method in the samples from the province of Sucumbíos and Orellana in eastern Ecuador were approximately three times higher than the antioxidant capacity found in Castanhal, state of Pará, in Brazil [58].

Figure 5 shows that the Açaí in Orellana (*Euterpe oleracea* 2) has a higher antioxidant activity than the Açaí cultivated in Sucumbíos (*Euterpe oleracea* 1). For the two methods, FRAP and ABTS, the Orellana species is 26.7% and 13.2% larger than the Sucumbíos species. The concentration of polyphenols and flavonoids in Orellana’s Açaí is approximately 36% higher than that of Sucumbíos (FRAP) and 15% (ABTS).

## 4. Discussion

The analyses carried out in this document have been carried out despite limitations such as the proximity between the regions and their climatic similarity; this could yield similar results that are complex to discuss comparatively. There was also a limitation in the number of samples obtained because Açaí production in the regions studied only occurs once a year. The results of this article may arouse interest in the Ecuadorian agro-industrial, pharmaceutical, and health sectors, which can take advantage of the massive production and consumption of Açaí (*Euterpe oleracea*) and open export markets for the products, ingredients, and functional bioactives obtained that are already appreciated around the world.

The determination of the physicochemical, nutritional, and antioxidant properties and antioxidant capacity of Açaí has shown that the state of maturity and the geographical location do influence the quantitative characteristics of the aforementioned parameters.

The increase in total solids concentration (TSS) and decrease in acidity observed in this study during the change in the maturity state from E1-10% to E3-100% (Table 1) can be explained by the effect of enzymatic activity (hydrogenases), consumption of the organic acids themselves in the synthesis of new components, and hydrolysis of starch accumulated before the maturation period as reported by [61,62].

The altitudes of the Sucumbíos and Orellana regions, where the Açaí samples were cultivated and collected, were 1102 and 403 m, respectively. The Açaí of Sucumbíos had a higher concentration of TSS than that of Orellana, possibly related to the higher altitude of Sucumbíos, demonstrating similar trends with the TSS/altitude relationship [63,64]. TSS concentration increases, and acidity decreases at higher altitudes, probably due to increased photosynthesis due to increased solar radiation and the conversion of organic acids into sugars through the process of gluconeogenesis, as reported in [65,66].

The fat content of the Açaí of the Sucumbíos region (altitude of 1000 m) was 28% higher than that found in Orellana (403 m). In [67,68,69], results with similar trends and a positive relationship between height and fat content were reported. Environmental factors at different altitudes influence the respiration of Açaí trees, affecting fruit metabolism during ripening and influencing the total amount of fat [70].

The fiber, fats, and proteins of the Açaí of the two regions studied increased as the fruit matured (see Table 2). At the same time, the concentration of carbohydrates decreased, presenting results similar to the studies reported in [71,72,73]; this behavior has been associated with ethylene that activates different processes in ripening, such as the synthesis of carotenoids and monounsaturated fats and the degradation of chlorophyll and carbohydrates; it is important to mention that the Açaí in the ripening process has climacteric fruit behavior [74,75].

It is observed that the content of PT in the Açaí is influenced by the degree of maturity and the geographical location of the crop (see Table 3); the concentration of polyphenols and flavonoids decreases while the degree of ripeness of the fruit increases. According to studies [76,77,78,79], results with the same trend have been reported in different fruits.

In the first phase of maturity (E1-10%), the initial concentration of phenolic compounds and flavonoids is higher in Orellana than in Sucumbíos; see Table 3. The Orellana area has an altitude and rainfall 63% and 36% lower than that of Sucumbíos. In [80,81], the same trends were reported by conducting a study on various fruits, confirming that rainfall and altitudes can inversely influence the total concentration of phenolic compounds and flavonoids.

The TCA concentration in this study showed an accentuated tendency to increase as fruit ripening occurred; in Table 3, it was observed that there are significant differences in TCA concentration in the three different ripening stages studied and differences in the regions, and this tendency was also observed in the Morus, Fragaria, Aronia, Prunus, Vaccinium, Rubus, and Vitis genera of berries and fruits [82,83,84,85]. These results suggest that anthocyanins are synthesized during the ripening process and that in red and purple fruits, there is an accumulation of TCA at the end of ripening, which gives these fruits their characteristic color since anthocyanins are responsible for the intense red to violet color of fruits according to [86,87,88,89]. The increase in the concentration of anthocyanins in ripe fruits could be due to the action of phenylpropanoids and the enzyme chalcone synthetase, which participate in the biosynthesis of anthocyanins [90]. Figure 6 shows the AT and PT values of the various South American countries that cultivate Açaí (*Euterpe oleracea*) [1,2,3,4,5,6], as well as the values obtained in this study in the regions of Sucumbíos and Orellana in Ecuador. It is observed that the concentrations of antioxidants in Açaí grown in Ecuador are similar in the amount of PT to those grown in Brazil and Venezuela; however, in Açaí grown in Ecuador, a high concentration of TA stands out, which enhances its nutritional potential and antioxidant bioactive characteristics that are highly appreciated in the international market. It has been shown that there is only one harvest period per year for Açaí in the regions shown in Figure 6, which occurs between the months of January and March.

The data provided in Table 4 and Table 5 report a higher antioxidant activity when there is a higher concentration of polyphenols and flavonoids; however, AA decreases at a lower concentration of anthocyanins, and similar trends were reported in [77,91,92,93]. The hydroxyl groups abundantly present in the polyphenolic groups and/or flavonoids are responsible for antioxidant activity [94], showing that the maturation process reduces antioxidant activity (Table 4 and Table 5).

In this study, the antioxidant activity and concentration of anthocyanins of Açaí in Orellana are higher than that of Sucumbíos, and the height of Orellana is 63% lower than that of Sucumbíos. The results of this study suggest that low altitude inversely influences anthocyanin concentrations and antioxidant activity, see Table 4 and Table 5, with similar trends reported in [95]. Other studies report that the concentration of anthocyanins is higher at higher altitudes [93,96,97] and report that climatic conditions can benefit or affect the AA parameter or phenol concentration depending on the variety of the crop. The modified and non-modified genomes of each crop in their various stages of development have different responses to climate, soil, and other conditions reported in [98,99].

## 5. Conclusions

The state of maturity and the cultivation region of the Açaí influence the increase in TSS content and the decrease in TA. The fiber, fats, and proteins of the Açaí increase as the fruit ripens, while the carbohydrate content decreases. Total polyphenols and flavonoids decrease as ripening increases, and anthocyanins decrease. There is greater antioxidant activity at a higher concentration of polyphenols and flavonoids; however, AA decreases when the concentration of anthocyanins is lower. The concentration of anthocyanins from Açaí grown in the Amazon region of Ecuador is a higher concentration than that found in Açaí cultivation in other South American countries, suggesting an untapped market potential.

The extensive number of products, ingredients, and bioactives that can be obtained from Açaí have the potential to contribute to and boost multiple sectors of the industry, such as pharmaceuticals, food, and health. Further studies on the profile of polyphenols and anthocyanins should be carried out to strategically exploit the identified bioactive components.

## Figures and Tables

**Figure 1 foods-13-03046-f001:**
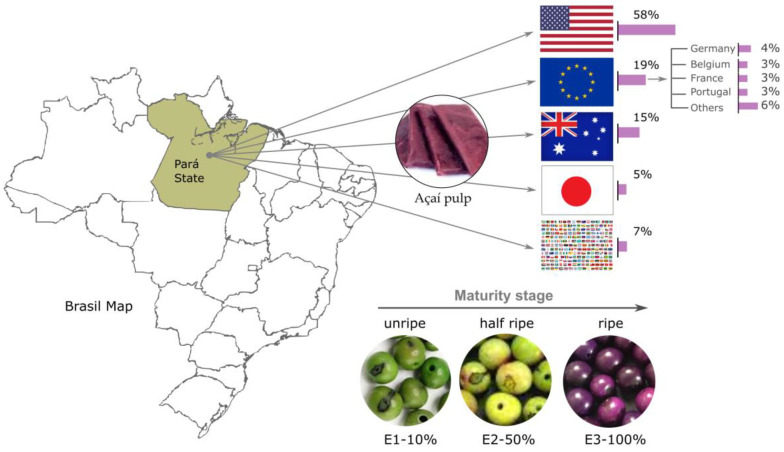
Export of Brazilian Açaí to different regions of the world.

**Figure 2 foods-13-03046-f002:**
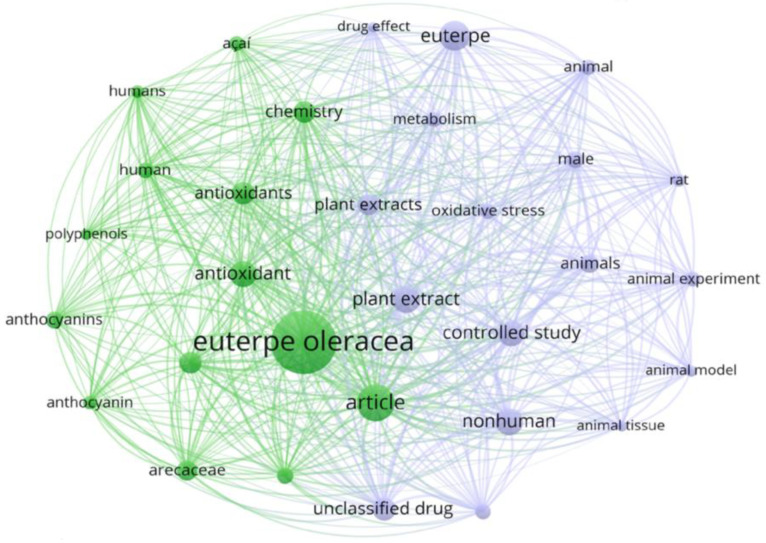
Relevant studies related to Açaí (*Euterpe oleracea*). Graph generated with VOSViewer® 1.6.20 version.

**Figure 3 foods-13-03046-f003:**
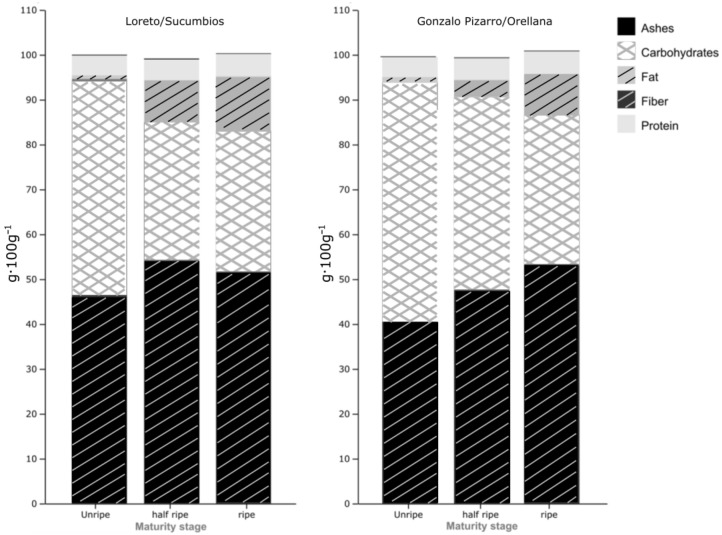
The concentration of nutritional parameters of Açaí cultivated in Ecuador in species obtained in Sucumbíos (*Euterpe oleracea* 1) and Orellana (*Euterpe oleracea* 2) in different stages of maturation.

**Figure 4 foods-13-03046-f004:**
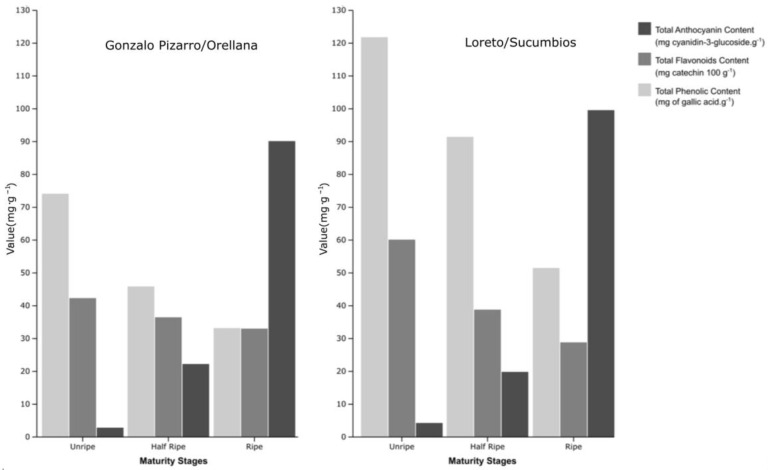
Effect of different harvest maturity stages and two geographical provinces (Orellana and Sucumbíos) in Ecuador on bioactive compounds. Total phenolic content, total flavonoid content, and total anthocyanin content of Açaí (*Euterpe oleracea*). Values are presented as mean ± SE, n = 3.

**Figure 5 foods-13-03046-f005:**
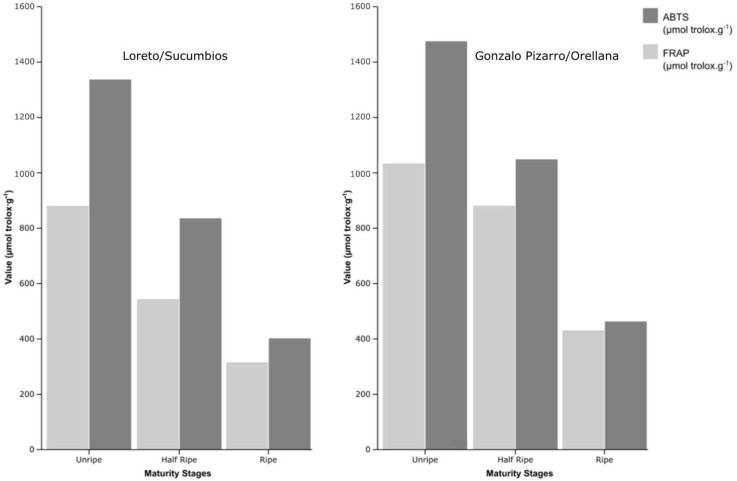
Effect of different harvest maturity stages and two different geographical regions (Sucumbíos and Orellana) in Ecuador on antioxidant activity measurement for ABTS, and FRAP radical scavenging activity of Açaí (*Euterpe oleracea*). Values are presented as mean ± SE, n = 3.

**Figure 6 foods-13-03046-f006:**
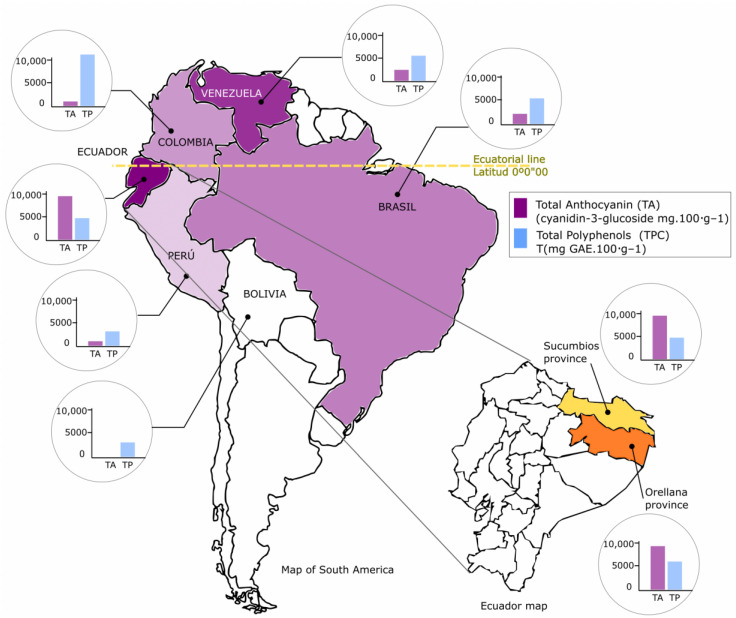
Concentration of anthocyanins (TAs) and total polyphenols (TPs) in various South American countries.

**Table 1 foods-13-03046-t001:** Açai Parameters of the regions of Sucumbíos and Orellana in Ecuador.

Parameters	Sucumbíos (Gonzalo Pizarro)	Orellana (Loreto)
Temperature (mean and gradient)	29.5–21.1	29.1–21.5
Average annual rainfall (mm)	3029.9–4816.4/average of 3923	2029.3–3029.9 (2529)
Relative humidity	74–84 (average of 79)	75–84 (79.5)
Hot days above 30 °C (4-year average)	145	116
Warm days around 20 °C (4-year average)	358	331
Average height (masl)	1102 m https://goo.su/nKdh accessed on 18 August 2024	403 m https://goo.su/nX3b accessed on 18 August 2024
Average evaporation (mm)	400–850 (average of 625)	400–1000 (average of 700)

**Table 2 foods-13-03046-t002:** Physicochemical analysis of Açaí in three stages of maturity and two production areas.

City/Province	Maturity Stage	Soluble Solids (°Brix)		Titratable Acidity (% Citric Acid)		Maturity Index (TSS/TA)	
Gonzalo Pizarro/Sucumbíos	Unripe	0.45 ± 0.07	Cf	0.44 ± 0.003	Ab	1.01 ± 0.15	Cd
	Half ripe	1.75 ± 0.07	Bc	0.44 ± 0.002	Ab	3.94 ± 0.17	Bc
	Ripe	3.6 ± 0.14	Aa	0.26 ± 0.011	Bc	13.97 ± 0.07	Ab
Loreto/Orellana	Unripe	0.65 ± 0.07	Ce	0.54 ± 0.003	Aa	1.2 ± 0.12	Cd
	Half ripe	1.45 ± 0.07	Bd	0.45 ± 0.004	Bb	3.24 ± 0.13	Bc
	Ripe	2.75 ± 0.07	Ab	0.22 ± 0.003	Cd	12.61 ± 0.50	Aa

The mean ± standard deviation (n = 3) expresses the results found. Significant differences between maturity stages (*p* < 0.05) are represented by capital letters, and significant differences between the two provinces (*p* < 0.05) are represented by lowercase letters, using ANOVA followed by Tukey’s test.

**Table 3 foods-13-03046-t003:** Proximal analysis of Açaí in three stages of maturity and two production areas.

City/Province	Maturity Stage	Ashes (g·100 g^−1^)		Protein (g·100 g^−1^)		Fat (g·100 g^−1^)		Fiber (g·100 g^−1^)		Total Carbohydrates (g·100 g^−1^)	
Gonzalo Pizarro/Sucumbíos	Unripe	0.15 ± 0.01	Ac	4.38 ± 0.58	Ab	0.83 ± 0.04	Ae	46.46 ± 0.26	Ae	48.18 ± 0.53	Ab
	Half ripe	0.18 ± 0.01	Aa	4.61 ± 0.19	Aab	9.38 ± 0.33	Ab	54.07 ± 0.03	Aa	30.92 ± 0.23	Ae
	Ripe	0.12 ± 0.01	Ab	4.95 ± 0.09	Aab	12.15 ± 0.11	Aa	51.55 ± 0.05	Ac	31.57 ± 0.01	Ae
Loreto/Orellana	Unripe	0.14 ± 0.01	Bbc	4.43 ± 0.17	Bab	1.27 ± 0.04	Be	40.51 ± 0.47	Bf	53.30 ± 0.27	Ba
	Half ripe	0.15 ± 0.01	Bc	4.78 ± 0.27	Bab	4.03 ± 0.10	Bd	47.51 ± 0.24	Bd	42.95 ± 0.57	Bc
	Ripe	0.13 ± 0.01	Bbc	5.36 ± 0.55	Ba	8.74 ± 0.18	Bc	53.10 ± 0.15	Bb	33.59 ± 0.17	Bd

Results expressed as the mean ± standard deviation (n = 3). Capital letters indicate significant differences between states of maturity (*p* < 0.05). Lowercase letters indicate significant differences between provinces (*p* < 0.05) using the ANOVA analysis followed by Tukey’s test.

**Table 4 foods-13-03046-t004:** Functional compounds in Açai in three stages of maturity and two cultivated areas.

City/Province	Coating Color	Total Polyphenols		Total Flavonoids		Total Anthocyanins	
		(PT)		(FT)		(ACT)	
		(mg Gallic Acid·g^−1^)		(mg (+)-Catechin·g^−1^)		(mg Cyanidin-3-O Glu·g^−1^)	
Gonzalo Pizarro/Sucumbíos	Unripe	74.14 ± 3.40	Ac	42.32 ± 1.29	Ab	2.86 ± 0.25	Ce
	Half ripe	45.87 ± 1.72	Be	36.47 ± 0.76	Bbc	22.25 ± 0.42	Bc
	Ripe	33.20 ± 2.18	Cf	33.03 ± 0.39	Ccd	90.16 ± 1.53	Ab
Loreto/Orellana	Unripe	121.81 ± 2.55	Aa	60.14 ± 2.28	Aa	4.28 ± 0.78	Ce
	Half ripe	91.45 ± 1.06	Bb	38.82 ± 4.12	Bbc	19.85 ± 0.33	Bd
	Ripe	51.50 ± 2.17	Cd	28.84 ± 3.19	Cd	99.59 ± 0.65	Aa

Results are expressed as the mean ± standard deviation (n = 3) on a dry basis. Capital letters indicate significant differences between states of maturity (*p* < 0.05). Lowercase letters indicate significant differences between provinces (*p* < 0.05) using the ANOVA analysis followed by Tukey’s test.

**Table 5 foods-13-03046-t005:** Antioxidant activity of Açaí from two cultivated areas in the Amazon region of Ecuador.

City/Province	Coating Color	Antioxidant Activity		Antioxidant Activity	
		(FRAP)		(ABTS)	
		(µmol trolox·g^−1^)		(µmol trolox·g^−1^)	
Gonzalo Pizarro/Sucumbíos	Unripe	880.58 ± 19.69	Ab	1336.96 ± 89.25	Ab
	Half ripe	544.10 ± 7.32	Bc	835.76 ± 6.96	Bd
	Ripe	315.43 ± 4.96	Ce	402.41 ± 10.38	Ce
Loreto/Orellana	Unripe	1033.87 ± 19.98	Aa	1474.97 ± 35.54	Aa
	Half ripe	881.42 ± 10.26	Bb	1048.63 ± 23.40	Bc
	Ripe	430.94 ± 9.23	Cd	463.22 ± 24.92	Ce

Results are expressed as the mean ± standard deviation (n = 3) on a dry basis. Capital letters indicate significant differences between states of maturity (*p* < 0.05). Lowercase letters indicate significant differences between provinces (*p* < 0.05) using the ANOVA analysis followed by Tukey’s test.

## Data Availability

The original contributions presented in the study are included in the article. Further inquiries can be directed to the corresponding author.

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
