# Peer review of "Physicochemical Characterization and Antioxidant Capacity of Açaí (Euterpe oleracea) in Ecuadorian Region"

_foods, 2024, doi:10.3390/foods13193046_

Round 1

Reviewer 1 Report

Comments and Suggestions for Authors

The topic is highly relevant as it addresses a significant gap in the literature regarding the characterization of Açaí grown in Ecuador, which has been less studied compared to Brazilian Açaí. However, several areas require revision. I recommend that the authors address these issues and resubmit the manuscript for further consideration.

Introduction: The introduction provides a global background on Açaí, but it lacks a clear statement of the research gap and specific objectives of the study. While the introduction mentions the lack of studies on Ecuadorian Açaí, it would be helpful to highlight the novelty of this research in comparison to other studies. Some of the references cited in the introduction are outdated; I suggest including more recent studies to provide a current perspective on the topic. Additionally, expanding the literature review to include studies on Açaí from different locations and countries would help underscore the novelty and importance of the current research. The introduction is primarily focused on describing the existence of Açaí varieties in different regions, without fully connecting this to the research presented.

Materials and Methods:

  • In the section describing sample collection, it would be helpful to include more information on the selection criteria for the sampling sites. Why were Sucumbíos and Orellana chosen? How representative are these regions of the overall Açaí production in Ecuador?
  • The profile of phenolic compounds was not determined. Do the authors believe this omission might limit the characterization of the samples concerning their functional characteristics and effects? As stated in the introduction (lines 77-79), “To market and produce Açaí, it is essential to carry out the phytochemical characterization, the concentration of antioxidants (polyphenols, flavonoids, and anthocyanins), and the antioxidant activity of the bioactive components of the cultivated species.”
  • What time of year were the harvests conducted? Could the timing of harvests have influenced the results?

Results: The visual representation of the results is critical. Figures and tables should be used strategically to complement the text. In particular, figures could be improved in terms of clarity, especially regarding labeling, scale, and color use. In my opinion, there is no need to divide section 3.1 into 3.1.1.

Discussion:

The discussion provides an interpretation of the results, but it would benefit from a more critical analysis. For instance, the manuscript could address potential limitations of the study in greater depth. This includes discussing the potential variability in Açaí composition due to environmental factors and limitations in the methodologies used. It would also be useful to compare the findings more extensively with those from other regions where Açaí is cultivated, to provide a more comprehensive understanding in a global context.

The discussion could be enhanced by exploring the practical implications of the findings, particularly in terms of their potential impact on the Açaí industry in Ecuador and possible applications in food and health products.

Conclusion: the conclusion should be more focused. While it summarizes the study well, it could be more impactful by highlighting the key findings and suggesting specific areas for future research, such as applications in the food and pharmaceutical industries.

Comments on the Quality of English Language

The manuscript is generally well-written, but there are some instances where the language could be refined. For example, some sentences are overly complex. Additionally, there are occasional grammatical errors and awkward phrasing. Please ensure the manuscript is proofread.

Author Response

Response to Reviewer 1 Comments

Dear Reviewer, we appreciate the time you have taken to complete this review. Your comments have greatly improved the quality of our article. Below are responses to the changes made in accordance with your comments. We have attached the details in .docx format.

Comments 1:   The introduction provides a global background on Açaí, but it lacks a clear statement of the research gap and specific objectives of the study.

Response 1: Lines 94-99 have been clarified to highlight the lack of nutritional characterization studies of the species Euterpe Oleracea cultivated in Ecuador. In addition, the objectives of this development are described.

“Currently, no scientific articles study the species Euterpe Oleracea cultivated in Ecuador, and this species has not been characterized or evaluated in terms of its nutritional and functional characteristics”. Los objetivos de este trabajo son determinar parámetros psicoquimicos, nutricionales, concentración de antioxidants y actividad antioxidante de la variedad de azai (Euterpe oleracea) en tres estados de maduración (E1-10%. E2-50%, and E3-100%) y dos localizaciones geográficas en el Ecuador (Sucumbíos and Orellana).

Comments 2:  While the introduction mentions the lack of studies on Ecuadorian Açaí, it would be helpful to highlight the novelty of this research in comparison to other studies.

Response 2: A description of the reasons why the article is novel has been incorporated in lines 99-102.

This work is novel since it provides for the first time reference data on the açaí (Euterpe oleracea) cultivated in Ecuador and serves as a basis for future studies of marketing feasibility and a better use of this fruit.

Comments 3: Some of the references cited in the introduction are outdated; I suggest including more recent studies to provide a current perspective on the topic.

Response 3: A clarification on the novelty of this work has been incorporated in lines 102-104.

In addition to this, the values obtained from Anthocyanins and Total Phenols for the same species in the countries of South America: Venezuela, Brazil, Colombia, Peru, Bolivia and Ecuador have been compared.

Comments 4: Additionally, expanding the literature review to include studies on Açaí from different locations and countries would help underscore the novelty and importance of the current research

Response 4: In addition to this, the values obtained from Anthocyanins and Total Phenols for the same species in the countries of South America: Venezuela, Brazil, Colombia, Peru, Bolivia and Ecuador have been compared.

Comments 5: The introduction is primarily focused on describing the existence of Açaí varieties in different regions, without fully connecting this to the research presented.

Response 5: In addition to this, the values obtained from Anthocyanins and Total Phenols for the same species in the countries of South America: Venezuela, Brazil, Colombia, Peru, Bolivia and Ecuador have been compared.

Comments 6: The introduction is primarily focused on describing the existence of Açaí varieties in different regions, without fully connecting this to the research presented.

Response 6: A clarification of the reason that has prompted the creation of this article for the Ecuadorian species has been included in lines 91-95.

Comments 7: In the section describing sample collection, it would be helpful to include more information on the selection criteria for the sampling sites. Why were Sucumbíos and Orellana chosen? How representative are these regions of the overall Açaí production in Ecuador?

Response 7: Some clarifications have been included in lines 120-128 on the reasons for choosing Sucumbios and Orellana, as well as the importance and representativeness of the region in the traditional use of Açaí.

The regions of Sucumbíos and Orellana were chosen for two fundamental criteria: (1) they are the regions where the Açaí species Euterpe Oleraceae is cultivated in Ecuador and (2) the climate of the region has characteristics similar to the regions in Brazil (humid tropical climate).

Sucumbíos and Orellana are representative regions because acaí is known and used in traditional practices from the ancestral knowledge of the natives of the region. In addition to this, the soil is rich and of great diversity in which species are preserved without alterations due to the presence of large neighboring cities in a climate that corresponds to a tropical ecosystem with abundant rainfall and warm temperature throughout the year.

 Comments 8:   The profile of phenolic compounds was not determined. Do the authors believe this omission might limit the characterization of the samples concerning their functional characteristics and effects? As stated in the introduction (lines 77-79), “To market and produce Açaí, it is essential to carry out the phytochemical characterization, the concentration of antioxidants (polyphenols, flavonoids, and anthocyanins), and the antioxidant activity of the bioactive components of the cultivated species”.

Response 8: We agree with your comment; in reality, we believe that our scientific contribution brings pioneering data regarding the cultivation of Acai in the two main potential production regions of Euterpe Oleracea in Ecuador and surely stimulates the national and international scientific community to fill the gaps that this work did not consider due to lack of time and resources. The complete profile of polyphenols is certainly a study that should be carried out among others.

Comments 9:   What time of year were the harvests conducted? Could the timing of harvests have influenced the results?

Response 9:   Samples of green and semi-ripe fruits were taken in November (dry period-low rainfall) and harvest of ripe fruits was carried out in January (rainy period). The species cultivated in these regions is characterized by having only one production per year. The discussion (lines 585-586) mentions the possible influence of the amount of rainfall on the production of polyphenols and total flavonoids, and the influence of the geographical height of the regions (lines 568-572) is also addressed. Possible influences on total anthocyanin levels and crop height are described (lines 619-623).

Comments 10. The visual representation of the results is critical. Figures and tables should be used strategically to complement the text. In particular, figures could be improved in terms of clarity, especially regarding labeling, scale, and color use. In my opinion, there is no need to divide section 3.1 into 3.1.1.

Response 10. Figure 3 has been modified, the visualization has been improved for better identification of the components. In addition to this, 3.1 and 3.1.1 have been added

Comments 11:   The discussion provides an interpretation of the results, but it would benefit from a more critical analysis. For instance, the manuscript could address potential limitations of the study in greater depth. This includes discussing the potential variability in Açaí composition due to environmental factors and limitations in the methodologies used. It would also be useful to compare the findings more extensively with those from other regions where Açaí is cultivated, to provide a more comprehensive understanding in a global context.

Response 11:   Comments on limitations have been included in the study (lines 553-557).

The analyses carried out in this document have been carried out despite limitations such as the proximity between the regions and their climatic similarity, this could lead to similar results that are complex to discuss comparatively. There was also a limitation in the number of samples obtained because acai production in the regions studied only occurs once a year.

The potential variability in the composition of açaí due to environmental factors has been commented on lines 623-635.

The data provided in Tables 4 and 5 report a higher antioxidant activity when there is a higher concentration of polyphenols and flavonoids; however, AA decreases at a lower concentration of anthocyanins, and similar trends were reported by [77, 91-93]. The hydroxyl groups abundantly present in the polyphenolic groups and/or flavonoids are responsible for antioxidant activity [94], showing that the maturation process reduces antioxidant activity (Tables 4 and 5).

In this study, the antioxidant activity and concentration of anthocyanins of Açaí de Orellana is higher than that of Sucumbíos, and the height of Orellana is 63% lower than that of Sucumbíos. The results of this study suggest that low altitude inversely influences anthocyanin concentrations and antioxidant activity see Tables 4 and 5, with similar trends reported in [95]. Other studies report that the concentration of anthocyanins is higher at higher altitudes [93,96-97] and report that climatic conditions can benefit or af-fect the AA parameter or phenol concentration depending on the variety of the crop.

The influence of environmental factors on the parameters studied in most of the online discussion in: (572-574), (580-582), (594-599), (633-635)

The concentration values of total polyphenols and antiocyanins have been addressed in a comparative manner (lines 33-44) in Figure 6 and described in lines 611-617.

Figure 6 shows the AT and PT values of the various South American countries that culti-vate Açaí (Euterpe Oleracea) [1-6], as well as the values obtained in this study in the regions of Sucumbíos and Orellana in Ecuador. It is observed that the concentrations of antioxi-dants in Açaí grown in Ecuador are similar in the amount of PT to those grown in Brazil and Venezuela; however, in Açaí grown in Ecuador, a high concentration of TA stands out, which enhances its nutritional potential and antioxidant bioactive characteristics that are highly appreciated in the international market.

Comments 12:   The conclusion should be more focused. While it summarizes the study well, it could be more impactful by highlighting the key findings and suggesting specific areas for future research, such as applications in the food and pharmaceutical industries.

Response 12:  

Suggestions on future work that may impact the Ecuadorian industry have been included in the conclusions section.

 The extensive number of products, ingredients, bioactives that can be obtained from acai, have the potential to contribute to and boost multiple sectors of the industry such as pharmaceuticals, food and health. Further studies on the profile of polyphenols and antiocyanins should be carried out to strategically exploit the bioactive components identified.

Response to Comments on the Quality of English Language

Point 1: The manuscript is generally well-written, but there are some instances where the language could be refined. For example, some sentences are overly complex. Additionally, there are occasional grammatical errors and awkward phrasing. Please ensure the manuscript is proofread.

Response 1: The full text, grammatical errors and expressions that were not clear to the reader have been reviewed.

Reviewer 2 Report

Comments and Suggestions for Authors

line 33, "Euterpe oleracea " italic

line 109, change 1000g  to 1 kg

lines 100-106, delete

line 111, 0.025816,-77.3619604 mean?

line 113,  E1, E2, E3: represent how many days after flowering?

line 194, The numbers in the chemical formula should be subscripted.

line 292,     700 nm?     Please verify that the experimental method in the manuscript is accurate.

line 316, talbe 1  0,45 ± 0,07 should change to 0.45± 0.07.  

table 1, Column  2 : Maturity stage.  use unripe, half ripe, ripe. The same was as below.

line 318, p<0.05 or p<0.01?

line 349, table 2, Column 4, 6 ,  Are these letters marked correctly? Please check other talbes.

line 373, The discrimination of each indicator in Figure 3 is not significant, please redraw.

line 398, mg Gallic acid.g-1,  “·” not “.”

line 460, talbe 4 can be adjusted to materials and methods.

line 472, Redraw Figure 4 and add standard deviation. And table 5.

line 589, Are the harvesting periods and standards consistent in different regions?

There should be a space between numbers and units.

Standardized references

Author Response

Response to Reviewer 2 Comments

Dear Reviewer, we appreciate the time you have taken to complete this review. Your comments have greatly improved the quality of our article. Below are the responses to the changes made in accordance with your comments. We have attached the details in .docx format.

Comments 1:   line 33, "Euterpe oleracea " italic.

Response 1: The change has been made (Euterpe oleracea)

Comments 2:  line 109, change 1000g  to 1 kg

Response 2:  The change has been made Samples of 1 kg of Açaí

Comments 3:   lines 100-106, delete

Response 3:  Text has removed

Section two refers to the materials, reagents, and samples used for the study and the methodologies used in the analyses. Section 3 presents the physicochemical characterization, nutritional quality, quantification of antioxidant compounds, and determination of the antioxidant capacity of the species Euterpe Oleracea (Açaí). In section 4, he discusses the findings of section 3 and compares the potential of the Açaí species cultivated in Ecuador with those cultivated in Brazil and other Latin American countries; finally, the conclusions are presented

Comments 4:   line 111, 0.025816,-77.3619604 mean?.

Response 4:  We have improved the description.

The coordinates of the sampling site are 0°02′49"N 77°19′22"W (Sucumbios) and 0°41′25"S 77°18′30"W (Orellana).

Comments 5:   line 113,  E1, E2, E3: represent how many days after flowering?

Response 5: E1, E2, E3 represent the three stages of maturation (E1: unripe, E2: half ripe , E3: ripe).

This is explained in lines 128-132.

Comments 6:   line 194, The numbers in the chemical formula should be subscripted.

Response 6: The change has been made.

(3.5g K2SO4 and 0.4g CuSO4X5H2O),

Comments 7:   line 292,     700 nm?     Please verify that the experimental method in the manuscript is accurate.

Response 7: The value of 734 nm described in reference 34 is confirmed.

Comments 8:   Table 1, Column 2 : Maturity stage.  use unripe, half ripe, ripe. The same was as below.

Response 8: Updated the table with the changes in all tables.

Comments 9:   line 318, p<0.05 or p<0.01?.

Response 9: It is confirmed that the value p<0.05

Comments 10:   line 349, table 2, Column 4, 6 ,  Are these letters marked correctly? Please check other tables.

Response 10: The letters are well written, according to the results of the ANOVA analysis and the comments below the tables for upper and lower case letters

Comments 11:   line 373, The discrimination of each indicator in Figure 3 is not significant, please redraw.

Response 11: The figure has been modified for a better understanding of the reader

Comments 12:   line 398, mg Gallic acid.g-1,  “·” not “.”.

Response 12: The change has been made

Comments 13:   line 460, table 4 can be adjusted to materials and methods.

Response 13: Table 4 has been changed to Materials and Methods

Comments 14:   line 472, Redraw Figure 4 and add standard deviation. And table 5.

Response 14: Figure 4 has been verified and redrawn according to the correct values in Table 4. In addition to this, the standard deviation has been added below table 5

Comments 15:   line 589, Are the harvesting periods and standards consistent in different regions?.

Response 15:  The collection of ripe Ecuadorian acai was carried out in January, in the countries of Figure 6 the collection was carried out between the months of February and April.

Comments 16: Should there be a space between numbers and unit regions?

Response 16:  Fixed the spacing between numbers and units

Comments 17:   Standardized references

Response 17:  The bibliography has been reviewed, updated, and improved

Round 2

Reviewer 1 Report

Comments and Suggestions for Authors

The authors have addressed the reviewers' suggestions satisfactorily. In my opinion, the article is now suitable for publication in the journal.